# Elastic Fibers and F-Box and WD-40 Domain-Containing Protein 2 in Bovine Periosteum and Blood Vessels

**DOI:** 10.3390/biomimetics8010007

**Published:** 2022-12-23

**Authors:** Mari Akiyama

**Affiliations:** Department of Biomaterials, Osaka Dental University, 8-1, Kuzuhahanazono-cho, Hirakata, Osaka 573-1121, Japan; mari@cc.osaka-dent.ac.jp; Tel.: +81-72-864-3056

**Keywords:** elastic fiber, F-box and WD-40 domain-containing protein 2, calcification, bovine periosteum, blood vessel

## Abstract

Elastic fibers form vessel walls, and elastic fiber calcification causes serious vascular diseases. Elastin is a well-known elastic fiber component; however, the insoluble nature of elastic fibers renders elastic fiber component analysis difficult. A previous study investigated F-box and WD-40 domain-containing protein 2 (FBXW2) in the cambium layer of bovine periosteum and hypothesized that fiber structures of FBXW2 are coated with osteocalcin during explant culture. Here, FBXW2 was expressed around some endothelial cells but not in all microvessels of the bovine periosteum. The author hypothesized that FBXW2 is expressed only in blood vessels with elastic fibers. Immunostaining and Elastica van Gieson staining indicated that FBXW2 was expressed in the same regions as elastic fibers and elastin in the cambium layer of the periosteum. Alpha-smooth muscle actin (αSMA) was expressed in microvessels and periosteum-derived cells. Immunostaining and observation of microvessels with serial sections revealed that osteocalcin was not expressed around blood vessels at 6 and 7 weeks. However, blood vessels and periosteum connoted elastic fibers, FBXW2, and αSMA. These findings are expected to clarify the processes involved in the calcification of elastic fibers in blood vessels.

## 1. Introduction

Calcification is essential for bone regeneration; however, vascular calcification with atherosclerosis, diabetes, chronic kidney disease, or Mönckeberg’s calcification can cause cardiovascular mortality, atherosclerosis, and vessel wall rupture [1,2,3]. Given its relationship with vascular calcification, the calcification of elastic fibers has been investigated [4,5,6,7,8]. Elastin and fibrillin-1 are the major components of elastic fibers [9,10,11,12,13]; however, because of the insoluble nature of elastic fibers, analysis of elastic fiber components is difficult, and numerous elastic fiber-associated proteins and their elastogenesis processes remain unclear [14].

The periosteum is divided into two parts: the fibrous layer and the cambium layer [15,16]. In a previous study, the author identified the expression of F-box and WD-40 domain-containing protein 2 (FBXW2) in the cambium layer of bovine periosteum [17]. FBXW2 has been found to localize with osteocalcin [18], and it may be related to the osteogenic differentiation of periosteum-derived cells (PDCs). Alpha-smooth muscle actin (αSMA) is a marker of myofibroblasts [19] and early osteoprogenitor cells in the periosteum [20].

A previous study utilizing an explant culture of bovine periosteum reported that both FBXW2 and osteocalcin have a fiber-like structure and that osteocalcin is located on FBXW2 [17]. This previous study demonstrated that FBXW2 is expressed around some endothelial cells but not in all microvessels. The author hypothesized that FBXW2 is expressed only in blood vessels with elastic fibers because some microvessels are composed of endothelial cells and do not have elastic fibers. FBXW2 is known for β-catenin ubiquitylation [21]. However, no study has investigated the relationship between FBXW2 and elastic fibers. Therefore, in this study, immunostaining of elastin and FBXW2 and Elastica van Gieson (EVG) staining were performed. Elastic fibers are involved in the cambium layer of the periosteum and blood vessel walls. The aims of this study were to (i) demonstrate that the expression pattern of FBXW2 is similar to that of elastic fibers; (ii) ascertain whether FBXW2 is associated with elastic fibers; and (iii) investigate whether the calcification of elastic fibers occurs in blood vessel walls in a manner similar to periosteum calcification during the formation of periosteum-derived cells in the course of bone regeneration.

## 2. Materials and Methods

### 2.1. Preparation of Histologic Samples of Blood Vessel, Periosteum, and Bone

Blood vessels (under the skin), periosteum, and bone tissues (with periosteum) were isolated from the legs of 30-month-old Japanese Black Cattle (Kobe Chuo Chikusan, Kobe, Japan). A part of the periosteum from 2 forelegs or 2 hindlegs was fixed, and the rest of the periosteum was used for explant culture. The legs of four 30-month-old cattle were used the next day after their sacrifice for beef products, and the use of periosteum for culture was approved by the Osaka Dental University Regulations on Animal Care and Use committee (Approval No. 21-02001). After fixing the tissues with 4% paraformaldehyde (PFA), paraffin-embedded blocks were prepared. As previously described, no living animals were used in the study. Therefore, ARRIVE Essential 20 was not applicable. The samples were from female or castrated male cows.

### 2.2. Primary Culture of Bovine Periosteum-Derived Cells with Periosteum

The separated periosteum was used for explant culture in Medium 199 (Gibco, Grand Island, NY, USA; 12340-030) with 10% fetal bovine serum (Biosera, Kansas City, MO, USA; FB-1365/500), penicillin/streptomycin (FUJIFILM Wako Pure Chemical Corporation, Osaka, Japan; 168-23191), and 5 mg/mL ascorbic acid (Sigma, St. Louis, MO, USA; A0278-25G), and maintained at 37 °C with 5% CO_2_. The medium was changed once per week. Complexes of PDCs and periosteum were removed from tissue culture dishes and fixed with 4% PFA after explant culture, and paraffin-embedded blocks were prepared for sectioning.

### 2.3. Histological and Immunohistochemical Analysis

To better examine the elastic fibers, the paraffin-embedded sections were stained with EVG stain. The following primary antibodies were used for fluorescent immunostaining and immunohistochemistry: Anti-Von Willebrand factor antibody (1:300, ab6994; Abcam, Cambridge, UK), anti-alpha-smooth muscle actin antibody (1:1000, ARG66381; Arigo Biolaboratories Corp., Hsinchu City, Taiwan), anti-elastin antibody (1:200, BA-4; sc-58756; Santa Cruz Biotechnology, Inc., Santa Cruz, CA, USA), anti-FBXW2 antibody (1:200, #PA5-18,189; Invitrogen, Eugene, OR, USA), and anti-osteocalcin monoclonal antibody (1:500, code no. M042, clone no. OCG2; Takara Bio Inc., Shiga, Japan). The antibodies for FBXW2 and osteocalcin were the same as those used in Akiyama’s study in 2021 [17]. For the anti-αSMA antibody, antigen retrieval was performed under heat-mediated conditions using Tris/EDTA buffer (pH 9.0, 40 min; Diagnostic Bio-Systems, Pleasanton, CA, USA). For all other primary antibodies, antigen retrieval was performed using Proteinase K (S3020: Dako Cytomation, Glostrup, Denmark) for 10 min. Rabbit primary antibodies used were Alexa Fluor™ 594 tagged anti-rabbit IgG (H+L) (A11037; Invitrogen) and Alexa Fluor™ 488 tagged anti-rabbit IgG (H+L) (A32731; Invitrogen). Alexa Fluor™ 488 tagged anti-mouse IgG (H+L) (A11029; Invitrogen) was used as the mouse primary antibody and anti-goat IgG-CFL 594 (sc-516243; Santa Cruz Biotechnology, Inc.) was used as the goat primary antibody. 4′,6-Diamidino2-phenylindole (DAPI; Dojindo Laboratories, Kumamoto, Japan) was used for cell nucleus staining. For microscopic observation of secondary antibodies, alkaline phosphatase-tagged Simple Stain AP (R) (H2103; Nichirei Biosciences Inc., Tokyo, Japan) and Simple Stain AP (M) (H2102; Nichirei Biosciences Inc.), and anti-goat IgG-AP (sc-2355; Santa Cruz Biotechnology, Inc.) were used. To visualize alkaline phosphatase-labeled antibodies, PermaRed/AP (K049; Diagnostic BioSystems, Pleasanton, CA, USA) and PermaBlue/AP (K058; Diagnostic Bio-Systems) were used. A microscope (BZ-9000; Keyence Japan, Osaka, Japan), BZ-II Viewer (Version1.1; Keyence), and BZ-II Analyzer software (Version1.1; Keyence) were used for all images except for low-magnification EVG imaging. System Microscope (BX53; Olympus, Tokyo, Japan), Digital Camera (DP74; Olympus), and imaging software cellSens (ver.3.2, Olympus) were used for low-magnification EVG imaging. Adobe Illustrator (Adobe, San Jose, CA, USA) was used for combination figures.

## 3. Results

Fluorescent immunostaining showed the presence of Von Willebrand factor (Figure 1a) and αSMA (Figure 1b) in microvessels of the bovine periosteum at day 0. Double fluorescent immunostaining showed that two types of microvessels were present: microvessels with FBXW2 (arrow) and those without FBXW2 (Figure 1c). As there were two types of microvessels (microvessels with elastic fibers and those without elastic fibers), I observed elastin, the main component of elastic fibers, to evaluate the relationship between FBXW2 and elastic fibers. Elastin was present in the microvessels of the periosteum (arrow in Figure 1d). Elastin and FBXW2 were abundant in the cambium layer of the periosteum, except in the microvessels (Figure 1c,d), and EVG imaging showed that FBXW2 was present at the location of the elastic fibers (Figure 1e–g).

Figure 2a–e show the findings of the histological analysis of blood vessels. Elastin and FBXW2 were present at the medial and adventitia regions of the blood vessels (Figure 2a–c). Elastin localized in the same region as FBXW2 in the tissue of the blood vessels. Fluorescent immunostaining showed that blood endothelial cells were supported by FBXW2 in the medial region (Figure 2c,d). However, the expression of osteocalcin was not observed (Figure 2e) and blood vessels were not calcified around the elastic fibers.

In Appendix A, the structural similarities between elastic fibers and FBXW2 are demonstrated. Appendix A presents images of EVG (left) and FBXW2 (right). The left and right images were samples from the same paraffin blocks, but neither is an image of a serial section. Appendix A shows high-magnification images of bones, indicating similar sizes, shapes, and expressed regions between the elastic fibers and FBXW2. Previously, FBXW2 was reported to be expressed in bones [17], but it was not compared to elastic fibers. Appendix A show the periosteum during an in vitro explant culture at 3–5 weeks. The images of the elastic fibers and FBXW2 in the explant culture reflected the similarity between the two. The regions in which elastic fibers were present had FBXW2 as well.

On day 0, αSMA expression was observed only in vascular smooth muscle cells in microvessels (Figure 1b). Figure 3a–e show fluorescent immunostaining of αSMA in the periosteum from 1 to 5 weeks. At 1 week, αSMA was present in vascular smooth muscle cells, and a few αSMA-positive periosteal cells were present in the periosteum (Figure 3a). From 2 to 5 weeks, the presence of αSMA-positive cells was confirmed in the periosteum, and the expression of αSMA was apparent in PDCs (Figure 3b–e). Before explant culture, αSMA was expressed only in vascular smooth muscle cells, but during explant culture, αSMA-positive cells were observed in the periosteum, PDCs, and microvessels (Figure 3a–e). Immunohistochemical analysis performed using alkaline phosphatase labeling revealed that the primary antibody for αSMA was unsuitable for double immunostaining (Appendix A). Single immunostaining of αSMA was apparent (Appendix A); however, αSMA immunostaining after using the osteocalcin antibody was obscure (Appendix A).

From 6 to 7 weeks, a comparison of EVG imaging (Figure 4a,e) and fluorescent immunostaining of FBXW2 (Figure 4b,f) and αSMA (Figure 4c,g) showed that elastic fibers and FBXW2 were present in the same regions in microvessels. A comparison of fluorescent immunostaining of FBXW2 and osteocalcin (Figure 4d,h) showed that osteocalcin was not expressed around a part of the microvessels.

## 4. Discussion

The similarity in staining between the elastic fibers and the fiber structure of FBXW2 tends to demonstrate that FBXW2 could be associated with elastic fibers (Appendix A). I previously reported that the cambium layer of the periosteum contains FBXW2 [17,18]. However, it was not known why FBXW2 showed a fiber-like structure. Figure 1a–c indicate the presence of microvessels and Figure 1c shows that FBXW2 is expressed in microvessels that are indicated with arrows but not in other microvessels. Figure 1d demonstrates that the expression pattern of elastin is similar to that of FBXW2. Figure 1e–g represent images of the periosteum, cambium layer, and bone, respectively. Elastin and FBXW2 were expressed in the same region in the elastic fibers. As shown in Figure 1 and Appendix A, FBXW2 was expressed in the elastic fiber-containing regions in bone and periosteum tissues. Double immunostaining of elastin leads to weak fluorescence signals at low magnification (Appendix A). Therefore, single immunostaining was mainly employed. Moreover, double immunostaining did not quench the signal completely (Appendix A). Different fibers with similar shapes, sizes, and regions, except for elastic fibers, might be present along with the elastic fibers. However, at high magnification, double immunostaining of elastin and FBXW2 revealed that the two proteins formed structural units of the fiber (Appendix A). Elastic fibers are insoluble, and analysis of their components is difficult. In this study, FBXW2 and elastin were observed in similar regions in blood vessels. Therefore, FBXW2 might be associated with elastic fibers of the blood vessels. A comparison of FBXW2 and elastin localization within the periosteum indicated that some microvessels (indicated with arrows in the figures) contained both FBXW2 and elastin, whereas microvessels without elastin did not contain FBXW2. In large blood vessels, the expression patterns of elastin and FBXW2 were similar. Both elastin and FBXW2 within elastic fibers might play a role in supporting the blood vessel structure. Zhou et al. [22] studied lung cancer with FBXW2 as a suppressor, whereas Ren et al. [23] studied breast cancer with FBXW2 as a suppressor. Their studies focused on ubiquitination by FBXW2. However, to my knowledge, this is the first study on elastic fibers and FBXW2. FBXW2 has mainly been observed intracellularly and is known to induce ubiquitination [21]; it is unclear how FBXW2 is secreted into the extracellular space. Previously, I used two antibodies against FBXW2 (one from Abcam [18] and another from Invitrogen [17]). Antibodies against the synthetic peptide with sequence KRGSSFLAGEHPG, corresponding to the C terminal amino acids 410–422 in humans, were used, but at different periods with different labeled (peroxidase, alkaline phosphatase, and fluorescence) secondary antibodies [17,18]. These antibodies revealed the presence of FBXW2 in the extracellular space. In this study, both FBXW2 and elastic fibers were present in the tissue before the explant culture; moreover, cultured cells may not secrete FBXW2. Future studies should investigate how FBXW2 is secreted in the extracellular space (possibly by transportation with microvesicles or via unconventional protein secretion). Calcification was not observed in the blood vessels of 30-month-old cows. Previously, calcification was examined using alizarin red and von Kossa staining techniques to detect calcium deposits [24,25]. However, osteocalcin, an osteogenic differentiation marker, was examined to detect the calcification of vascular smooth muscle cells or vascular tissue [26,27,28,29,30]. In 2015, Hirashima et al. [31] reported that perforating fibers are present in the cambial layer, but the components of perforating fibers are still unclear. Elastic fibers may serve as the anchoring structure. The presence of FBXW2 in the periosteum and the relationship between FBXW2 and osteocalcin have been reported [18]; however, this study is the first to report the relationship between FBXW2 and elastic fibers.

αSMA is expressed in vascular smooth muscle cells and myofibroblasts [32,33]. In this study, I observed that as αSMA-positive cells proliferated, the multi-layer of bovine PDCs became thick. αSMA might function as a native scaffold in PDCs. Previously, Uematsu et al. [34] compared the control medium (Medium 199) with MesenPRO-RS™ medium and reported that αSMA was expressed on the surface areas of periosteal sheets expanded in MesenPRO, but not on those cultured in the control medium. Notably, in the present study, the multi-layer of bovine PDCs cultured in Medium 199 was found to express αSMA. The observed differences may be because Uematsu et al. investigated periosteum from human alveolar bones, and the age of the subjects was not clear; whereas, in this study, I investigated periosteum from 30-month-old bovine legs.

The primary antibody for αSMA was not suitable for double immunostaining (Appendix A); therefore, single immunostaining using serial sections was performed for αSMA. Antigen retrieval for αSMA requires heat-mediated conditions, whereas other antibodies require the use of proteinase K. Owing to the differences in the antigen retrieval conditions, double immunostaining of αSMA resulted in weak expression.

In this study, I also sought to determine whether calcification occurs around elastic fibers of microvessels in the periosteum. Elastic fibers are also contained in blood vessels. Under normal conditions, calcification of blood vessels rarely occurs. Elastic fibers in blood vessels do not contain osteocalcin (Figure 2e). Using double immunostaining, Akiyama [17] reported that FBXW2 and osteocalcin are co-expressed up to 7 weeks after explant culture. However, in this study, osteocalcin expression was not observed around small microvessels after explant culture for 6 and 7 weeks. Akiyama [18] reported a synthesized osteocalcin coat around the cord-like structure of FBXW2 during explant culture in the probable cambium layer of the periosteum. In this study, FBXW2 was expressed at the location of elastic fibers in the cambium layer of the periosteum and microvessels, whereas osteocalcin was expressed in the cambium layer, but not in microvessels. The different processes underlying the calcification of elastic fibers in the blood vessels and the cambium layer of the periosteum are still unknown. The periosteum contains osteogenic stem cells, whereas blood vessels do not. However, both tissues contain elastic fibers, FBXW2, and αSMA. A possible explanation is that FBXW2 in the cambium layer of the periosteum was uncovered, whereas FBXW2 in blood vessels was covered with endothelial cells and extracellular matrix, which made coating elastic fibers with osteocalcin difficult. FBXW2 was associated with elastin around elastic fibers in the cambium layer of the periosteum both before and during explant culture. Notably, Akiyama [18] reported solid fibers of FBXW2 and hollow fibers of osteocalcin. This finding can be explained by the fact that FBXW2 was present along with the elastic fibers and osteocalcin was expressed on these fibers.

This study had some limitations. As the periosteum does not contain large blood vessels, the calcification processes of large blood vessels were not investigated in this study. Unlike the periosteum, the large blood vessels could not be placed on the culture dishes. Therefore, the calcification of large blood vessels remains to be elucidated. In this study, I used one condition of culture using Medium 199. In the outgrowth of PDCs and osteocalcin, differences between forelegs and hindlegs were not observed. Periosteum from the cranium or jaw was not tested. Akiyama [17] reported that osteocalcin synthesis occurs in FBXW2 fibers and hypothesized that the co-expression of osteocalcin and FBXW2 is related to osteogenic differentiation and bone formation. Therefore, during the explant culture of the periosteum, calcification for bone regeneration (such as osteocalcin expression) might occur around elastic fibers.

In conclusion, the calcification of elastic fibers in microvessels was not observed. However, at the same time, elastic fibers in the cambium layer calcified. To my knowledge, this study is the first to clarify the process of intramembranous ossification through elastic fibers of the cambium layer. Further studies on the roles of FBXW2 and elastic fibers may provide insights into and solutions to prevent the calcification of blood vessels.

## Figures and Tables

**Figure 1 biomimetics-08-00007-f001:**
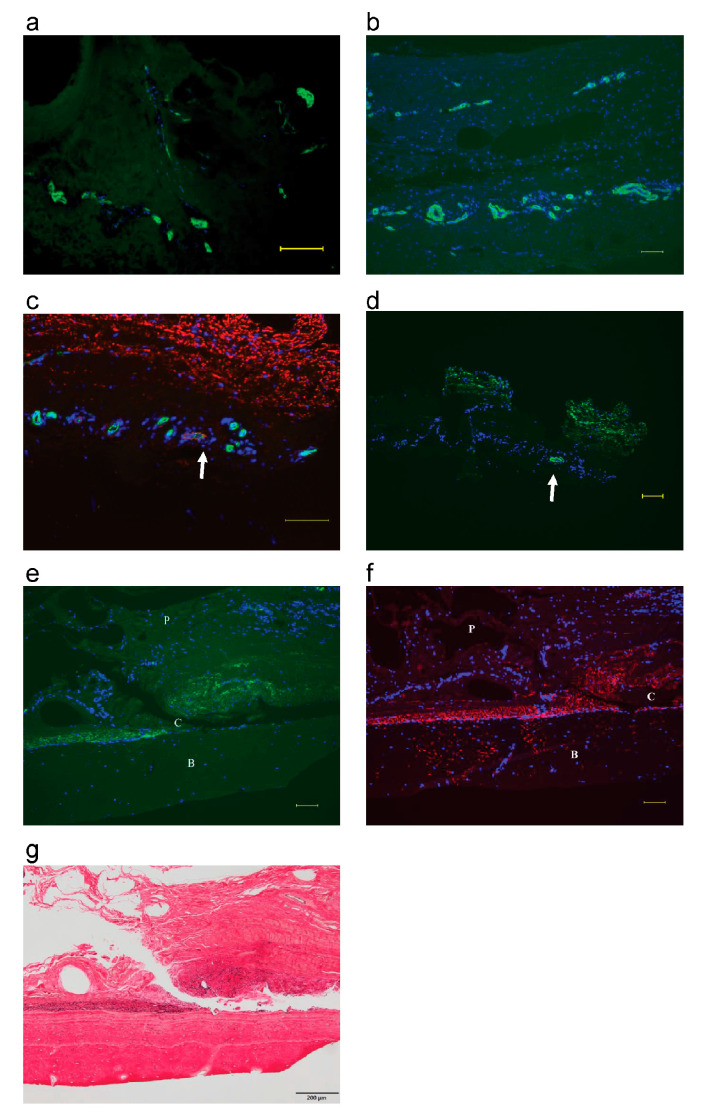
Localization of FBXW2 expression matched that of elastic fibers. (**a**–**d**) Fluorescent immunostaining of the periosteum on day 0. Scale bar: 100 μm: (**a**) Von Willebrand factor: green, 4′,6-diamidino2-phenylindole (DAPI): blue; (**b**) Alpha-smooth muscle actin (αSMA): green, DAPI: blue; (**c**) FBXW2: red, Von Willebrand factor: green, DAPI: blue, arrow: microvessel with FBXW2; (**d**) Elastin: green, DAPI: blue, arrow: microvessel. (**e**,**f**) Histology of the periosteum and bone on day 0. P: periosteum, C: cambium layer, B: bone. Scale bar: 100 μm; (**e**) Elastin: green, DAPI: blue; (**f**) FBXW2: red, DAPI: blue. For the image, the antibody used was the same as that used in Akiyama’s study (2021); (**g**) Elastica van Gieson (EVG) staining. Scale bar: 200 μm.

**Figure 2 biomimetics-08-00007-f002:**
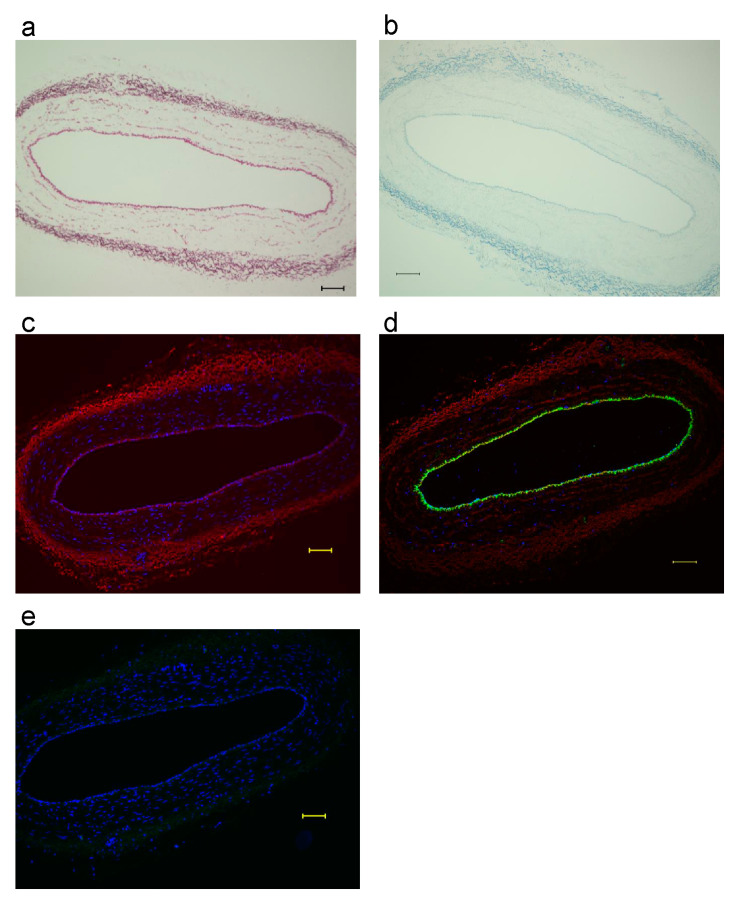
Histology of blood vessels: (**a**) Elastin: red; (**b**) FBXW2: blue; (**c**) FBXW2: red, DAPI: blue; (**d**) FBXW2: red, Von Willebranfactor: green, DAPI: blue; (**e**) Osteocalcin: green, DAPI: blue. Scale bar: 100 μm.

**Figure 3 biomimetics-08-00007-f003:**
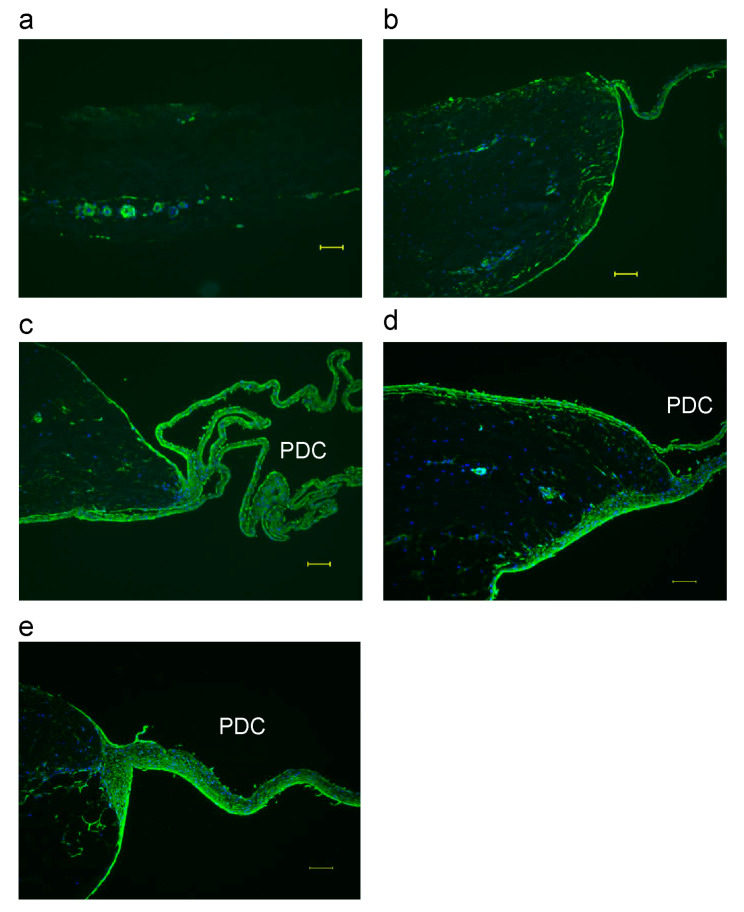
αSMA-positive cells increased during explant culture: (**a**–**e**) Fluorescent immunostaining of the periosteum. αSMA: green, DAPI: blue. PDC: Periosteum-derived cells at (**a**) 1 week; (**b**) 2 weeks; (**c**) 3 weeks; (**d**) 4 weeks; and (**e**) 5 weeks. Scale bar: 100 μm.

**Figure 4 biomimetics-08-00007-f004:**
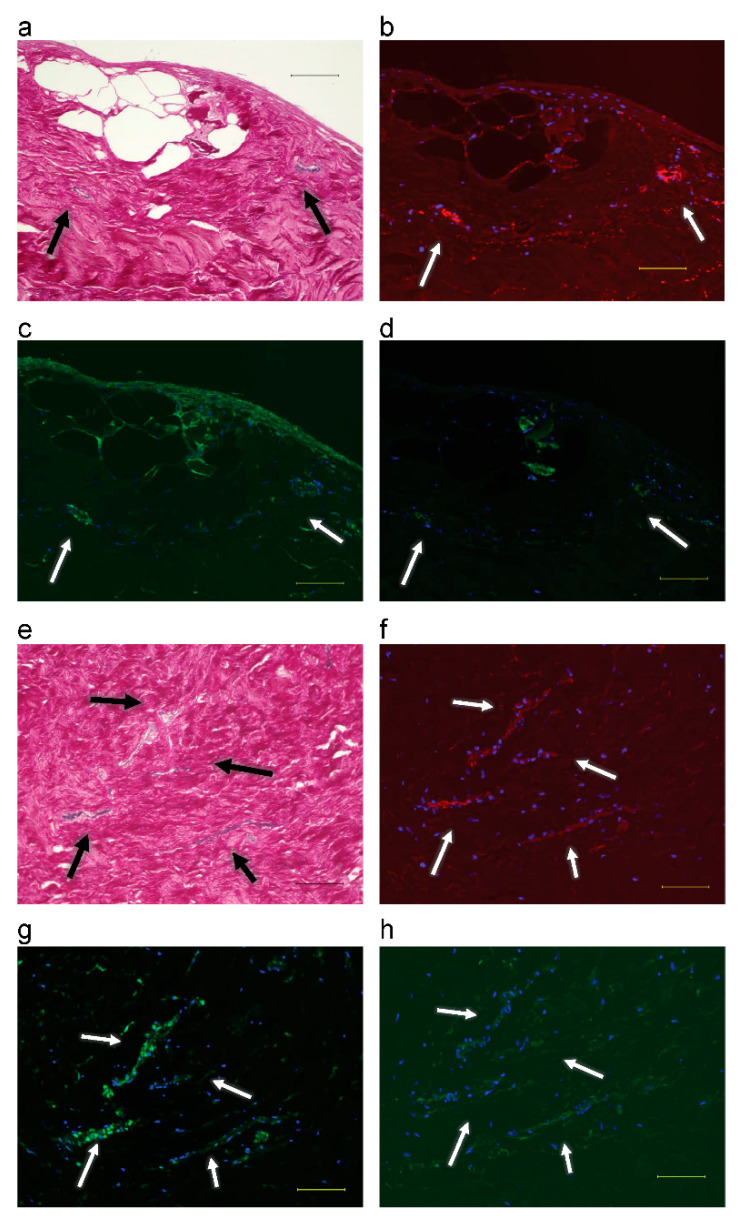
Observation of microvessels in serial sections of periosteum after explant culture: (**a**–**d**) At 6 weeks. Scale bar: 100 μm. (**a**) EVG staining; (**b**) FBXW2: red, DAPI: blue; (**c**) αSMA: green, DAPI: blue; (**d**) Osteocalcin: green, DAPI: blue. (**e**–**h**) At 7 weeks. Scale bar: 100 μm; (**e**) EVG staining; (**f**) FBXW2: red, DAPI: blue; (**g**) αSMA: green, DAPI: blue; (**h**) Osteocalcin: green, DAPI: blue. Arrows: blood vessels.

## Data Availability

The manuscript includes a fluorescent immunostaining image using the anti-FBXW2 antibody (Figure 1f). The same antibody was used in a previous study (Additional file 2: Appendix A; available at https://doi.org/10.1186/s13104-021-05825-z (Akiyama, accessed on 4 November 2021)), but samples from different cows were stained using this antibody.

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
