# Peer review of "Elastic Fibers and F-Box and WD-40 Domain-Containing Protein 2 in Bovine Periosteum and Blood Vessels"

_biomimetics, 2022, doi:10.3390/biomimetics8010007_

Round 1

Reviewer 1 Report

The study by Mari Akiyama relies on the investigation of FBXW2 in a bovine periosteum explant model preserved in culture. Immunohistofluorescence and classical histology staining were performed on serial sections or at different time points for 5 weeks. The results show quite clearly a strong correlation between the presence of elastic fibers and FBXW2 in small blood vessels and the cambium, although the role of this correlation is not obvious in the process of ectopic calcification.

-        Major concern:

So far, FBXW2 has mainly been observed intracellularly with a role in protein ubiquitination and proteasome addressing. FBXW2 does not contain a signal peptide to allow its secretion in the extracellular space, however, transport of FBXW2 by microvesicles remains possible, but needs to be demonstrated. If this protein were actually associated with elastic fibers (in the extracellular matrix), this would constitute an original discovery in the field. Nevertheless, the correlative data provided in this study are not sufficient to affirm this allegation. A confocal microscopy approach would undoubtedly be more convincing and resolving to observe a clear association of these two entities. The presence of FBXW2 within the elastic fibers could also be clearly demonstrated by immunogold labeling and transmission electron microscopy.

-        Minor concern:

Concerning the process of ectopic calcification, the author addresses this question exclusively through the presence of osteocalcin. Insofar, as histology is at the heart of the methodology, staining with alizarin red would be relevant to highlight calcium deposits within the sample.

Author Response

Thank you for giving me the opportunity to submit a revised draft of my manuscript. I appreciate the time and effort that you and the reviewers have dedicated to providing your valuable feedback on my manuscript. I am grateful to the reviewers for their insightful comments on my paper. I have been able to incorporate changes to reflect most of the suggestions provided by the reviewers. I have highlighted the changes within the manuscript. 

Here is a point-by-point response to the reviewers’ comments and concerns. My responses are shown in blue font.

Reviewer 2 Report

1.     Page 1, line 13: "I" is an extremely informal and colloquial word, it may not be appropriate for use in formal scientific writing.

2.     Page 7, line 176. "elastic fibers containing FBXW2 were observed" was mentioned, but without enough evidence.

3.     "In large blood vessels, the expression patterns of elastin and FBXW2 were similar " Page 7, line 176. This sentence should be explained more clearly and where does this conclusion come from?

4.     What is the meaning of " I investigated whether FBXW2 is a component of elastic fibers, similar to the cambium layer, and if so, whether calcification of elastic fibers in blood vessel walls occurs " (Page 2, lines 49 & 50) and "Figure 1e-g shows that the cambium layer also contains elastic fibers and elastin, which is a main component of elastic fibers" (Page 6, lines 167 & 168)? Please expand on the statements.

Author Response

(The authors gave the same response as above.)

Round 2

Reviewer 1 Report

I would like to thank the author for bringing new elements that further support the hypothesis. However, I would be very cautious that FBXW2 is a component of elastic fibers. The results demonstrate a potential co-localization, without providing evidence of a real association. I would therefore change the discussion sentence on page 7 "The similarity between the elastic fibers and the fibrous structure of FBXW2 indicates that FBXW2 is a component of the elastic fibers" by "The similarity between the staining of the elastic fibers and the fibrilar structure of FBXW2 tends to demonstrate that FBXW2 could be associated with elastic fibers".

Minor point:

- Fig S3 legend: "Elastin: blue" instead of "Elastin: bule"

Author Response

Response to reviewer

Comments: I would like to thank the author for bringing new elements that further support the hypothesis. However, I would be very cautious that FBXW2 is a component of elastic fibers. The results demonstrate a potential co-localization, without providing evidence of a real association. I would therefore change the discussion sentence on page 7 "The similarity between the elastic fibers and the fibrous structure of FBXW2 indicates that FBXW2 is a component of the elastic fibers" by "The similarity between the staining of the elastic fibers and the fibrilar structure of FBXW2 tends to demonstrate that FBXW2 could be associated with elastic fibers".

Minor point: - Fig S3 legend: "Elastin: blue" instead of "Elastin: bule"

Response: Thank you for your valuable comments. Per your suggestion, I have deleted the sentences indicating that elastic fibers contain FBXW2 or that FBXW2 is a component of elastic fibers. The following are the major revisions made in the manuscript:

Title: Elastic fibers and F-box/WD-40 domain-containing protein 2 in bovine periosteum and blood vessels

Page 2, lines 49–51:  The aims of this study were to (i) demonstrate that the expression pattern of FBXW2 is similar to that of elastic fibers; (ii) ascertain whether FBXW2 is associated with elastic fibers, …

Page 7, lines 175–177: The similarity in staining between the elastic fibers and the fiber structure of FBXW2 tends to demonstrate that FBXW2 could be associated with elastic fibers (Figures S1, S3).

Page 8, lines 189–191: However, at a high magnification, double immunostaining of elastin and FBXW2 revealed that the two proteins formed structural units of the fiber (Figure S3b–d).

Page 8, lines 205–208: Herein, I have discussed the antibodies used against FBXW2.

Page 9, lines 254–258: FBXW2 was associated with elastin around elastic fibers in the cambium layer of the periosteum both before and during explant culture. Notably, Akiyama [18] reported solid fibers of FBXW2 and hollow fibers of osteocalcin. This finding can be explained by the fact that FBXW2 was present along with the elastic fibers and osteocalcin was expressed on these fibers.

Page 9, line 273-275: Further studies on the roles of FBXW2 and elastic fibers may provide insights into and solutions to prevent the calcification of blood vessels.

Fig S3 legend: "Elastin: bule" has been corrected to "Elastin: blue."
